# Flood Frequency Analysis Using Participatory GIS and Rainfall Data for Two Stations in Narok Town, Kenya

**Eva Audrey Yessito Houessou-Dossou [1],\*** **, John Mwangi Gathenya [2], Mugwima Njuguna [3] and Zachary Abiero Gariy [4]**

1   Civil Engineering Department, Pan African University-Institute for Basics Sciences, Technology and Innovation/Jomo Kenyatta University of Agriculture and Technology, P.O. Box 62000-00200 Nairobi, Kenya

2   Soil, Water and Environmental Engineering Department/Jomo Kenyatta University of Agriculture & Technology, P.O. Box 62000-00200 Nairobi, Kenya; mgathenya@gmail.com

3   Centre for Urban Studies, School of Architecture & Building Sciences/Jomo Kenyatta University of Agriculture & Technology, P.O. Box 62000-00200 Nairobi, Kenya; mugwima@sabs.jkuat.ac.ke

4   School of Civil, Environmental & Geospatial Engineering/Jomo Kenyatta University of Agriculture & Technology P.O. Box 62000-00200 Nairobi, Kenya; zagariy@yahoo.co.uk

\*   Correspondence: houessoudossoueva@yahoo.com; Tel.: +254-743607745 or +229-66434697

**Abstract:** Flood management requires in-depth computational modelling through assessment of flood return period and river flow data in order to effectively analyze catchment response. The participatory geographic information system (PGIS) is a tool which is increasingly used for collecting data and decision making on environmental issues. This study sought to determine the return periods of major floods that happened in Narok Town, Kenya, using rainfall frequency analysis and PGIS. For this purpose, a number of statistical distribution functions were applied to daily rainfall data from two stations: Narok water supply (WS) station and Narok meteorological station (MS). The first station has a dataset of thirty years and the second one has a dataset of fifty-nine (59) years. The parameters obtained from the Kolmogorov–Smirnov (K–S) test and chi-square test helped to select the appropriate distribution. The best-fitted distribution for WS station were Gumbel L-moment, Pareto L-moment, and Weibull distribution for maximum one day, two days, and three days rainfall, respectively. However, the best-fitted distribution was found to be generalized extreme value L-moment, Gumbel and gamma distribution for maximum one day, two days, and three days, respectively for the meteorological station data. Each of the selected best-fitted distribution was used to compute the corresponding rainfall intensity for 5, 10, 25, 50, and 100 years return period, as well as the return period of the significant flood that happened in the town. The January 1993 flood was found to have a return period of six years, while the April 2013, March 2013, and April 2015 floods had a return period of one year each. This study helped to establish the return period of major flood events that occurred in Narok, and highlights the importance of population in disaster management. The study's results would be useful in developing flood hazard maps of Narok Town for different return periods.

**Keywords:** return period; goodness-of-fit; distribution; flood frequency; Narok Town

---

## 1. Introduction

Return period is an essential tool in hydrology that is used to estimate the time interval between events of a similar size or intensity. However, estimating the return period of such events can become an arduous task due to the fact of various reasons such as missing data, short times data series, or the

unknown probability distribution function of annual peaks [1]. Hence, frequency analysis is used to estimate the return period of specific events. This method of analysis can be used in the following among other applications, design of dams, bridges, culverts, and storm drainage channels. Frequency analysis can also be used in predicting the frequency of drought, in agricultural planning, as well as in flood prediction. Oosterbaan [1] named three methods of frequency analysis which are: interval method; ranking method; and applying theoretical frequency distributions. The last method involves the use of statistical distribution function to fit the data. Peak discharges are used for flood frequency analysis, but in the absence of a long record of discharge gauge data for any watershed, rainfall data series is used [2]. One of the reasons why rainfall data are more likely to be used for flood frequency analysis is that rainfall data are stochastic [3]. Several statistical distributions can be applied to the rainfall data. However, standard probability distribution functions commonly used in water resources engineering include normal, log-normal, Pearson, log Pearson type III and extreme value Type 1 (EVI) [4] distribution. Since there is nothing inherent in the series to indicate whether one distribution is more likely to be appropriate than another [5], goodness-of-fit test, which indicates how much the considered distributions fit the available data [6] is used to select the appropriate distribution. For instance, knowing that a major flood may be due to the accumulation of rain in the soil over several consecutive days, analysis of successive days of maximum rainfall can become an important tool for use in economic planning and for the design of hydraulic structures [2]. Such an analysis has been done by several researchers [2,7,8]. This study used the PGIS approach to spatial analysis in collecting data on past flood events and to identify major flood events. The PGIS uses geographic information systems to involve people in decision-making, and involves qualitative (population interview, observation) and quantitative (measurement) approach. The PGIS has been used in different sectors: indigenous land mapping [9,10]; flood management [11–13]; for conflicts resolution [14]; and natural resources management [15]. This study used a combination of two distinct approaches—PGIS to analyse past flood events and rainfall frequency analysis to determine the best-fitted distribution for the yearly rainfall to aid in computing the return period of the major flood event, with the ultimate goal of assessing the return period of major flood events in Narok town.

## 2. Materials and Methods

### 2.1. Study Area

The study area was Narok Town, which is the headquarters of Narok County, which is situated in the southwestern part of Kenya. Narok County borders Nakuru County to the North, Kajiado County to the East, Republic of Tanzania to the South, and Bomet, Kisii, Migori, and Nyamira counties to the West (See location Figure 1).

Narok catchment is formed by the Kakia and Esamburumbur subcatchment. The area of the watershed is 46.2 km$^2$, with the elevation varying from 1844 m to 2138 m. The main permanent river, Enkare Narok, passes through Narok town. However, Kakia and Esamburmbur dry valleys, which fall within the study area, often turn into rivers during heavy rains. The longest flow paths for Kakia and Esamburumbur measure 13.20 km and 10.01 km in length respectively, with an average slope of 18%. The downstream area experiences frequent flash flooding which results in harmful consequences in the town, such as loss of life, and destruction of property. Flooding also interferes with the local community's culture, threatens lives and livelihoods, and often result in decline in people's economic fortunes and poverty, among other negative effects. The topography of Narok town gives the town a basin-like formation, where floods drain through during the heavy rains. In the higher areas of the town, deforestation and inadequate drainage structures lead to flooding and cause road and buildings submersion, while the catchment is made up of agricultural land with maize and wheat being the most dominant crops. For the purposes of this study, two time series of rainfall data were collected: one at the water supply station (WS) and the other at the meteorological station (MS). The location of the two stations is shown in Figure 1. The rainfall data ranged from 1959 to 2018 for the MS, and from 1968

to 2018 for the WS. In the rainfall data, months with more than 30 days of missing values, especially for the rainy season (March to May; November, December) for a specific year, were not used in the analysis. Thus, the total useful data for WS station were 30 years, and the ones for MS were 59 years of data series.

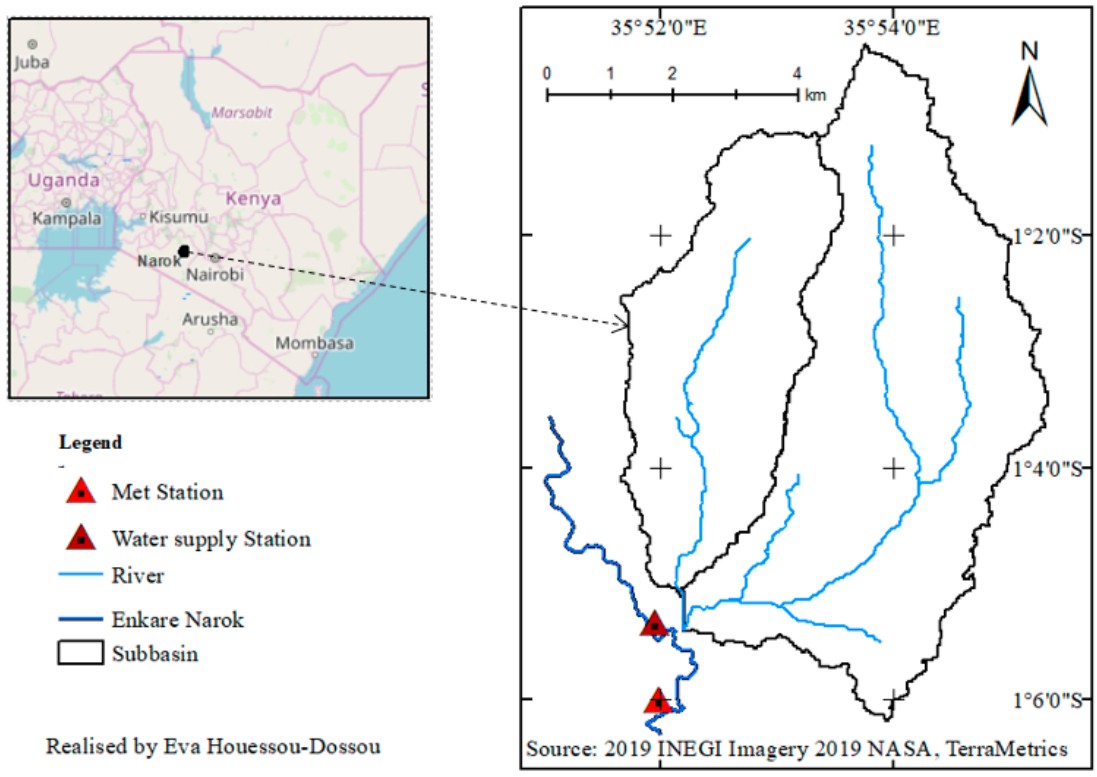

**Figure 1.** Location map.

## 2.2. PGIS

The National Centre of Geographic Information defines GIS as a system of hardware, software, and procedures to facilitate the management, manipulation, analysis, modelling, representation, and display of georeferenced data to solve complex problems regarding planning and management of resources. As a result of this definition, five functions are assigned to GIS, namely, data entry, data display, data management, retrieval, and information analysis [16]. Throughout recent decades, GIS has been used for environmental threat assessment. However, GIS hardware, software, and data are expensive and require a high level of technical expertise [14]. In addition, traditional GIS has been accused of not adequately addressing and incorporating social issues [17], which necessitated the inclusion of the term "Participatory GIS". Different researchers have examined the need to integrate the participation of the population in decision making, and the adequate means to achieve it. McCall [18] emphasized the need for precision in PGIS and stated that the degree of accuracy depends on the purpose of the PGIS. Despite the questions that have existed around participatory GIS, the method has been improved over the years and is increasingly used. Corbett et al. [19] applied PGIS for the assessment of social and ecological variation in Mpumalanga province, South Africa. The map obtained was based on people's knowledge. Tripathi and Bhattarya [20] evaluated the importance of integrating indigenous knowledge in GIS approach. The authors emphasized the importance of the participation of the local community in decision making. Rinner and Bird [21] used an online discussion forum for evaluating local community engagement in the development projects. The PGIS uses the diversity of experiences associated with "participatory development" [14] by involving people in GIS data collection and analysis in their community. For instance, a participatory approach to flood risk management requires the collection of information from the communities actually affected

by the flooding [13]. Depending on the availability of data, researchers either engage directly with the community or use already existing information on the community [13]. However, PGIS is a continuously evolving method and researchers keep discovering new ways in which the method can be applied in solving different problems. The method is continuously being used in adding to current information, finding out new and unknown information, alternative competing positions, discovering and interpreting people's "natural geography" [18]. The information collected using PGIS from the population in Narok, although incomplete (because the specific days of flooding could not be identified in some cases), proved beneficial for this study.

### 2.3. Flood Frequency Analysis

Generally, the steps followed in flood frequency analysis are as follows:

**Step 1**: Selection of the data

Here, annual maxima daily, annual maxima of two cumulative days and three cumulative days are selected for the analysis. This research focused on two and three days of rainfall because three-days rain flood records form an accurate representation of the magnitude of the flood flows [22]. In addition, the three-days rain flood discharge is the most critical duration for designing and evaluating flood mitigation [23].

**Step 2**: Fitting the probability distribution

Development of software for statistical extreme values analysis has been rapid [24]. Nowadays, different program package and pre-defined excel sheets are used to perform frequency analysis. Among the most used ones include: RAINBOW (developed by the Institute for Land and Water Management of the Katholieke Universiteit Leuven [25]); PeakFQ (that performs statistical flood-frequency analyses of annual-maximum peak flows [26]); CumFreq (initially developed for the analysis of hydrological measurements of variable magnitudes in space and time); Hydrognomon [27] developed by the ITIA (The name "Itia" is not an acronym, it is the Greek name of the willow tree) research team of National Technical University of Athens in 1997; and Hyfran (developed in Canada by the team of Bernard Bobée, chairman of statistical hydrology from 1992 to 2004). The Hyfran software can be used for any dataset of extreme values, provided that observations are independent and identically distributed [28]. Using frequently used probability functions such as normal, log-normal, Weibull, gamma, Gumbel, exponential or Pareto distribution, all these software can perform statistical analysis. This study used Hydrognomon software for rainfall frequency analysis. Hydrognomon is a software tool for the processing of hydrological data [27]. Although Hydrognomon is not commonly used in reviewed literature for flood frequency analysis, it was, nonetheless, used successfully by researchers to simulate the hydrology of Kaduna River in Niger [29], and for modelling future climatic variation [30]. Few researchers used Hydrognomon for time-series data analysis [29,30]. However, the software is freely available and can perform frequency analysis among many other hydrologic tasks. One of the advantages of this software is that it supports several time steps, from the most exceptional minute scales up to decades [27]; and filling of missing values. The software can also perform over thirteen statistical distributions and statistical test.

**Step 3:** Goodness of fit test to identify the best fitting distribution

Goodness-of-fit test statistics are used for checking the validity of a specified or assumed probability distribution model [31]. A goodness-of-fit test, in general, refers to measuring how well the observed data correspond to the fitted (assumed) model. The commonly used goodness-of-fit tests are Kolmogorov–Smirnov (K-S); root mean square error (RMSE) test, Chi-square test, and Anderson–Darling (A-D). The K–S test is an exact test where the distribution of the K–S test statistic itself does not depend on the underlying cumulative distribution function being tested. In addition, the use of Chi-square helps to understand the results and, thus, to derive more detailed information from the statistic test than from many others [32]. It also has the advantage in that it can be applied to any univariate distribution. For those reasons, the K–S test and Chi-square test have

been selected for testing the best-fitted distribution. The fact that Chi-square test requires a significant sample size is not a problem in the current study since we have a large sample size.

## 3. Results and Discussion

This study tested different distribution functions in order to compute the return period of significant flood events in Narok Town. Unfortunately, there is no meteorological station in the catchment. Although a Trans-African HydroMeteorological Observatory (TAHMO) station was installed in November 2018 in the northeastern part of the catchment, it lacked sufficient data for the analysis. However, the rainfall data from the MS and WS stations made it possible to confirm the information gathered during population interviews. Table 1 below presents the maximum daily precipitation for both WS and MS. Unfortunately, WS record has some missing data (from 1996 to 2010). In hydrology, missing data can lead to misunderstanding of rainfall variability and historical patterns [33]. The handling of missing data in meteorological time series is a relevant issue to many climatologic analyses [34]. The missing data can be filled using diverse techniques such as interpolation [35], correlation analysis [22] among adjacent stations, regression-based interval filling method [36], or inverse distance weighted techniques [37]. However, filling missing data can severely compromise its value for specific purposes [38]. Therefore, Oosterbaan [35] recommends that additional information (information used to fill data) be omitted from statistical analysis [38,39].

**Table 1.** Maximum daily rainfall of MS and WS station.

| Year | Max Daily P in mm | | Year | Max Daily P in mm | | Year | Max Daily P in mm | |
|------|------|------|------|------|------|------|------|------|
| | MS | WS | | MS | WS | | MS | WS |
| 1959 | 67.3 | - | 1979 | 41.7 | 75.1 | 1999 | 56.5 | - |
| 1960 | 69.9 | - | 1980 | 46.2 | 47.5 | 2000 | 68.2 | - |
| 1961 | 61.2 | - | 1981 | 30.3 | 101 | 2001 | 35.2 | - |
| 1962 | 46.2 | - | 1982 | 41.6 | 41.6 | 2002 | 68.8 | - |
| 1963 | 60 | - | 1983 | 40.7 | 41.1 | 2003 | 42 | - |
| 1964 | 61.6 | - | 1984 | 65.6 | 72 | 2004 | 78.2 | - |
| 1965 | 39.9 | - | 1985 | 70.3 | 58.5 | 2005 | 33.1 | - |
| 1966 | 39.1 | - | 1986 | 54.4 | 48.7 | 2006 | 59.4 | - |
| 1967 | 44 | - | 1987 | 51.5 | 56.4 | 2007 | 68.6 | - |
| 1968 | 63 | 72.3 | 1988 | 61.2 | 98 | 2008 | 49.1 | - |
| 1969 | 40.3 | 42.3 | 1989 | 54.2 | 65.5 | 2009 | 45.8 | - |
| 1970 | 75 | 50.5 | 1990 | 46.5 | 17.3 | 2010 | 43.1 | - |
| 1971 | 43.6 | 47.3 | 1991 | 29.2 | 29.3 | 2011 | 46.5 | 55.5 |
| 1972 | 55 | 44.5 | 1992 | 33.8 | 28.7 | 2012 | 74 | 67.4 |
| 1973 | 55.7 | 41.4 | 1993 | 81.7 | 101.5 | 2013 | 36 | 60.5 |
| 1974 | 78.2 | 71.8 | 1994 | 33 | 39 | 2014 | 55.6 | 54.8 |
| 1975 | 29.6 | 29.7 | 1995 | 81.4 | 79.5 | 2015 | 67 | 67 |
| 1976 | 38.2 | 91 | 1996 | 50.2 | - | 2016 | 52.8 | 50.5 |
| 1977 | 48.9 | 99.3 | 1997 | 139 | - | 2017 | 44.5 | 59.2 |
| 1978 | 101.2 | 101.2 | 1998 | 45.5 | - | | | |

Notably, the maximum daily rainfall per year varies among the stations (Figure 2). The values recorded for the amount of rainfall in the two stations were at times very close or similar as was the case of the year 1975 and 2015, where the recorded max rainfall was 29.6 and 67 mm for MS and 29.7

and 67 mm for WS station. The MS recorded the highest amount of rainfall in more years because it is located further to the South of WS station.

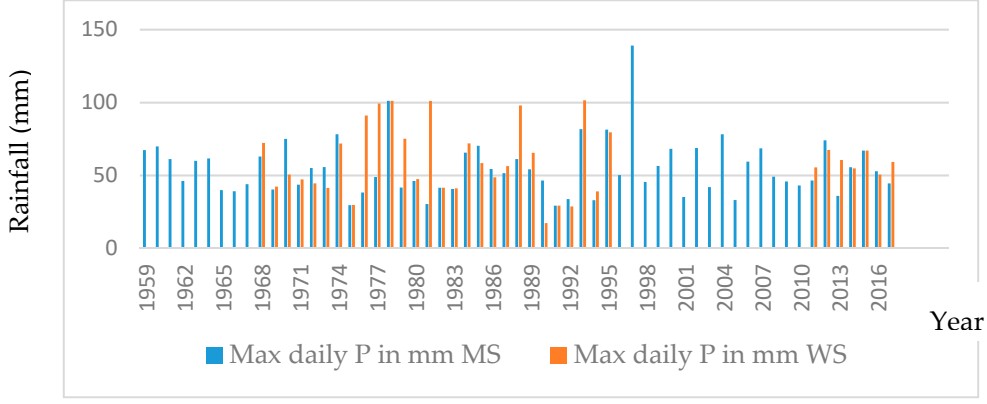

**Figure 2.** Max daily rainfall distribution for MS rainfall data (in blue) and WS data (in red).

### 3.1. Selection of the Best-Fitted Distribution

The maximum daily, maximum sum of two consecutive days, and the maximum sum of three consecutive day's rainfall of each year were used for statistical analysis. Tables 2 and 3 summarized the goodness-of-fit test results for each station. In the tables, LP III, Par, GEV, EV1, Gal, Gam, Pear, and LN represent log-Pearson type 3, Pareto, generalised extreme values, extreme values type 1, Galton, gamma, Pearson and log-normal distributions respectively.

**Table 2.** Summary of best-fitted distribution for water supply station.

| Water Supply Station | | | | | |
|---|---|---|---|---|---|
| **Max 1 Day** | | **Max 2 Days** | | **Max 3 Days** | |
| **K–S** | **Chi-Square** | **K–S** | **Chi-Square** | **K–S** | **Chi-Square** |
| LP III (0.08) | LP III (0.33) | Gal (0.09) | Gal (4.33) | Norm L- mom (0.07) | Norm L- mom (0.33) |
| Par (0.08) | Par (2.33) | Pear III (0.09) | Pear III (4.33) | **EV3 (0.07)** | **EV3 (0.07)** |
| GEV L-mom (0.08) | GEV L-mom (2.33) | Gam (0.09) | Gam (4.00) | GEV (0.07) | GEV (1.00) |
| **EV1 L-mom (0.08)** | **EV1 L-mom (0.33)** | GEV (0.09) | GEV (3.33) | GEV L-mom (0.07) | GEV L-mom (1.00) |
| Par (0.08) | Par (2.33) | GEV L-mom (0.08) | GEV L-mom (3.00) | EV3 L-mom (0.07) | EV3 L-mom (1.00) |
| GEV L-mom k specified (0.09) | GEV L-mom k specified (0.33) | **Par L-mom (0.07)** | **Par L-mom (1.7)** | Par L-mom (0.07) | Par L-mom (0.67) |

**Table 3.** Summary of best-fitted distribution for meteorological station.

| Meteorological Station | | | | | |
|---|---|---|---|---|---|
| **Max 1 Day** | | **Max 2 Days** | | **Max 3 Days** | |
| **k–s** | **Chi-Square** | **k–s** | **Chi-Square** | **k–s** | **Chi-Square** |
| LN (0.06) | LN (0.92) | LN (0.06) | LN (4.71) | LN (O.06) | LN (2.34) |
| Gal (0.05) | Gal (3.29) | Gal (0.06) | Gal (06.14) | **Gam (0.06)** | **Gam (1.62)** |
| LP III (O.05) | LP III (2.10) | LP III (0.06) | LP III (5.19) | LP III (0.05) | LP III (3.53) |
| EV1 (0.06) | EV1 (1.15) | **EV1 (0.06)** | **EV1 (4.71)** | EV1 (0.06) | EV1 (3.29) |
| GEV (0.06) | GEV (3.05) | GEV (0.06) | GEV (6.14) | EV1 L-mom (0.05) | EV1 L-mom (2.10) |
| **GEV L-mom (0.04)** | **GEV L-mom (1.15)** | GEV L-mom (0.06) | GEV L-mom (4.71) | GEV L-mom (0.05) | GEV L-mom (2.10) |

In the above tables (Tables 2 and 3), the selected distributions, which correspond to the one with the lowest value of K–S and Chi-square, are highlighted.

Goodness-of-fit test, as well as the skewness, standard deviation and the mean of the statistical parameters, were calculated. Water supply station was moderately skewed with a skewness equal

to 0.527 while the meteorological station was highly skewed, posting a skewness of 1.725, which is greater than 1. The statistical comparison by K–S and Chi-square test for goodness-of-fit showed that the best-fitted distribution for the water supply station were, respectively, extreme values L-moment (0.08, 0.33), Pareto L-moments (0.07, 1.7), and Weibull distribution (0.07, 0.07) for maximum daily, maximum two days and maximum three days rainfall data. For MS, the best-fitted distributions were generalized extreme value L-moment (0.04, 1.15), Gumbel (0.06, 4.71), and gamma (0.05, 2.1) distribution for maximum one day, two days, and three days, respectively. However, the selected distribution did not necessarily exclude the use of other methods. In fact, for WS station, GEV L-moments kappa specified could be used for maximum daily rainfall analysis, while GEV L-moment could be used for maximum two days rainfall and normal L-moment or Pareto L-moment for maximum three days rainfall. Otherwise, normal distribution L-moment was as good as the Weibull distribution for maximum three days rainfall analysis, since it showed results that were very close to Weibull distribution. For MS, log-normal distribution could also be appropriated for the maximum daily rainfall as well as the two days and the three days maximum rainfall.

### 3.2. 5, 10, 25, 50, and 100 Years Return Period Calculation

For the best-fitted distribution, Gumbel distribution was selected for return period calculation. Due to the small size of the data, 50 and 100 years return period intensity was computed using 95% confidence interval. Table 4 presents a summary of the 5, 10, 25, 50, and 100 years return period rainfall amount for each station, depending on the maximum daily rainfall or the maximum two days rainfall or three days.

**Table 4.** Summary table of 5, 10, 25, 50, and 100 years return period and corresponding rainfall intensity.

| WS | | MS | |
|---|---|---|---|
| **Max 1 Day** | | **Max 1 Day** | |
| **Gumbel L-moment** | | **GEV Min L-moment** | |
| **Return period (year)** | **Max daily Rainfall (mm)** | **Return period (year)** | **Max daily Rainfall (mm)** |
| 5 | 78.8022 | 5 | 68.8304 |
| 10 | 92.8842 | 10 | 80.0689 |
| 25 | 110.677 | 25 | 93.4172 |
| 50 | 123.876 | 50 | 102.734 |
| 100 | 131.592 | 100 | 111.55 |
| **Max 2 Days** | | **Max 2 Days** | |
| **Pareto L-moment** | | **Gumbel Max** | |
| **Return period (year)** | **Rainfall (mm)** | **Return period (year)** | **Rainfall (mm)** |
| 5 | 106.007 | 5 | 88.5849 |
| 10 | 118.705 | 10 | 101.557 |
| 25 | 128.513 | 25 | 117.948 |
| 50 | 132.789 | 50 | 130.107 |
| 100 | 135.464 | 100 | 142.177 |
| **Max 3 Days** | | **Max 3 Days** | |
| **EV3-Min (Weibull)** | | **gamma** | |
| **Return period (year)** | **Rainfall (mm)** | **Return period (year)** | **Rainfall (mm)** |
| 5 | 113.632 | 5 | 101.115 |
| 10 | 125.974 | 10 | 114.296 |
| 25 | 138.729 | 25 | 129.526 |
| 50 | 146.741 | 50 | 140.024 |
| 100 | 153.797 | 100 | 149.916 |

### 3.3. The Return Period of the Significant Flood Event

From population interview, reliable information was collected on significant flood events that occurred in the town. Additional information collected from the Water Resource Authority (WRA) helped as support. Some inhabitants were able to indicate the level where water reached on walls or trees. In spite of difficulties such as language barrier, age of the inhabitant (too young to remember the event) faced in some areas, it was possible to gather similar information on three major past floods in the town. Most of the inhabitants confirmed that significant flood events that occurred in the town happened on January 1993, April and May 2013, and 28th April 2015.

Most of the information recorded from the population interviewed was confirmed by the amount of rainfall recorded. However, in some cases, the amount of daily rainfall seemed insufficient to justify the occurrence of flood, necessitating the consideration of two or three days' rainfall. For example, in 2015 the maximum daily rainfall was recorded on 6th May. According to the interview, this did not lead to flooding. However, the summation of three days rainfall from 26th to 28th of April 2015 (maximum three days rainfall) which was 63.2 mm (Table 5) led to a flood. There is a need to notice that the month of April recorded almost daily rainfall. Thus, the accumulation of water in the ground and the excess of surface runoff, compounded by the sparce vegetation in the town could explain the occurrence of that flood on 28th April 2015.

**Table 5.** Rainfall distribution on the major flood events.

| | MS Rainfall in mm | | | WS Rainfall in mm | | | Date | MS Rainfall in mm | | | WS Rainfall in mm | | |
|---|---|---|---|---|---|---|---|---|---|---|---|---|---|
| Date | One Day | 2 Days | 3 Days | One Day | 2 Days | 3 Days | | One Day | 2 Days | 3 Days | One Day | 2 Days | 3 Days |
| 1/1/1993 | 0.0 | 0.0 | 0.0 | 0 | 0 | 0 | 5/1/2013 | 0.0 | 3.6 | 5.3 | 5 | 5 | 5 |
| 1/2/1993 | 0.1 | 0.1 | 0.1 | 0 | 0 | 0 | 5/2/2013 | 31.9 | 31.9 | 35.5 | 0 | 5 | 5 |
| 1/3/1993 | 0.0 | 0.1 | 0.1 | 0 | 0 | 0 | 5/3/2013 | 0.0 | 31.9 | 31.9 | 37 | 37 | 42 |
| 1/4/1993 | 2.1 | 2.1 | 2.2 | 1.9 | 1.9 | 1.9 | 5/4/2013 | 0.0 | 0.0 | 31.9 | 0 | 37 | 37 |
| 1/5/1993 | 0.0 | 2.1 | 2.1 | 0 | 1.9 | 1.9 | 5/5/2013 | 0.0 | 0.0 | 0.0 | 0 | 0 | 37 |
| 1/6/1993 | 0.0 | 0.0 | 2.1 | 0 | 0 | 1.9 | 5/6/2013 | 4.3 | 4.3 | 4.3 | 4.6 | 4.6 | 4.6 |
| 1/7/1993 | 0.6 | 0.6 | 0.6 | 0.9 | 0.9 | 0.9 | 5/7/2013 | 0.8 | 5.1 | 5.1 | 0 | 4.6 | 4.6 |
| 1/8/1993 | 5.8 | 6.4 | 6.4 | 5.7 | 6.6 | 6.6 | 5/8/2013 | 2.7 | 3.5 | 7.8 | 0 | 0 | 4.6 |
| 1/9/1993 | 1.4 | 7.2 | 7.8 | 1.9 | 7.6 | 8.5 | 5/9/2013 | 5.3 | 8.0 | 8.8 | 10.6 | 10.6 | 10.6 |
| 1/10/1993 | 0.0 | 1.4 | 7.2 | 0 | 1.9 | 7.6 | 5/10/2013 | 7.3 | 12.6 | 15.3 | 9.1 | 19.7 | 19.7 |
| 1/11/1993 | 0.0 | 0.0 | 1.4 | 0 | 0 | 1.9 | 5/11/2013 | 0.0 | 7.3 | 12.6 | 0 | 9.1 | 19.7 |
| 1/12/1993 | 21.7 | 21.7 | 21.7 | 22.7 | 22.7 | 22.7 | 5/12/2013 | 0.0 | 0.0 | 7.3 | 0 | 0 | 9.1 |
| 1/13/1993 | 8.4 | 30.1 | 30.1 | 8.1 | 30.8 | 30.8 | 5/13/2013 | 0.0 | 0.0 | 0.0 | 0 | 0 | 0 |
| 1/14/1993 | 6.7 | 15.1 | 36.8 | 12.8 | 20.9 | 43.6 | 5/14/2013 | 0.0 | 0.0 | 0.0 | 0 | 0 | 0 |
| 1/15/1993 | 2.5 | 9.2 | 17.6 | 3.3 | 16.1 | 24.2 | 5/15/2013 | 0.0 | 0.0 | 0.0 | 0 | 0 | 0 |
| 1/16/1993 | 39.7 | 42.2 | 48.9 | 40.8 | 44.1 | 56.9 | 5/16/2013 | 0.0 | 0.0 | 0.0 | 0 | 0 | 0 |
| 1/17/1993 | 10.8 | 50.5 | 53.0 | 5.3 | 46.1 | 49.4 | 5/17/2013 | 0.0 | 0.0 | 0.0 | 0 | 0 | 0 |
| 1/18/1993 | 0.0 | 10.8 | 50.5 | 0 | 5.3 | 46.1 | 5/18/2013 | 0.0 | 0.0 | 0.0 | 0 | 0 | 0 |
| 1/19/1993 | 26.6 | 26.6 | 37.4 | 27.5 | 27.5 | 32.8 | 5/19/2013 | 0.0 | 0.0 | 0.0 | 0 | 0 | 0 |
| **1/20/1993** | **81.7** | **108.3** | **108.3** | **101.5** | **129** | **129** | 5/20/2013 | 0.0 | 0.0 | 0.0 | 0 | 0 | 0 |
| 1/21/1993 | 0.0 | 81.7 | 108.3 | 0 | 101.5 | 129 | 5/21/2013 | 0.0 | 0.0 | 0.0 | 0 | 0 | 0 |
| 1/22/1993 | 2.7 | 2.7 | 84.4 | 33.1 | 33.1 | 134.6 | 5/22/2013 | 0.0 | 0.0 | 0.0 | 0 | 0 | 0 |
| 1/23/1993 | 29.1 | 31.8 | 31.8 | 0 | 33.1 | 33.1 | 5/23/2013 | 0.0 | 0.0 | 0.0 | 0 | 0 | 0 |
| 1/24/1993 | 0.0 | 29.1 | 31.8 | 0 | 0 | 33.1 | 5/24/2013 | 0.0 | 0.0 | 0.0 | 0 | 0 | 0 |
| 1/25/1993 | 0.0 | 0.0 | 29.1 | 0 | 0 | 0 | 5/25/2013 | 0.0 | 0.0 | 0.0 | 0 | 0 | 0 |
| 1/26/1993 | 0.0 | 0.0 | 0.0 | 0 | 0 | 0 | 5/26/2013 | 0.0 | 0.0 | 0.0 | 0 | 0 | 0 |
| 1/27/1993 | 0.6 | 0.6 | 0.6 | 0 | 0 | 0 | 5/27/2013 | 0.0 | 0.0 | 0.0 | 0 | 0 | 0 |
| 1/28/1993 | 0.8 | 1.4 | 1.4 | 1.4 | 1.4 | 1.4 | 5/28/2013 | 0.0 | 0.0 | 0.0 | 0 | 0 | 0 |
| 1/29/1993 | 0.0 | 0.8 | 1.4 | 0 | 1.4 | 1.4 | 5/29/2013 | 0.0 | 0.0 | 0.0 | 0 | 0 | 0 |
| 1/30/1993 | 16.9 | 16.9 | 17.7 | 14.2 | 14.2 | 15.6 | 5/30/2013 | 0.0 | 0.0 | 0.0 | 0 | 0 | 0 |
| 1/31/1993 | 0.0 | 16.9 | 16.9 | 0.5 | 14.7 | 14.7 | 5/31/2013 | 0.0 | 0.0 | 0.0 | 0 | 0 | 0 |
| 3/1/2013 | 0.0 | 0.0 | 0.0 | 0 | 0 | 0 | 4/1/2015 | 0.0 | 0.0 | 0.0 | 0 | 0 | 0 |
| 3/2/2013 | 0.0 | 0.0 | 0.0 | 0 | 0 | 0 | 4/2/2015 | 0.0 | 0.0 | 0.0 | 0 | 0 | 0 |
| 3/3/2013 | 0.0 | 0.0 | 0.0 | 0 | 0 | 0 | 4/3/2015 | 8.0 | 8.0 | 8.0 | 0 | 0 | 0 |

**Table 5.** *Cont.*

| Date | MS Rainfall in mm | | | WS Rainfall in mm | | | Date | MS Rainfall in mm | | | WS Rainfall in mm | | |
|---|---|---|---|---|---|---|---|---|---|---|---|---|---|
| | One Day | 2 Days | 3 Days | One Day | 2 Days | 3 Days | | One Day | 2 Days | 3 Days | One Day | 2 Days | 3 Days |
| 4/3/2013 | 0.0 | 0.0 | 0.0 | 0 | 0 | 0 | 4/4/2015 | 0.0 | 8.0 | 8.0 | 8 | 8 | 8 |
| 3/5/2013 | 0.0 | 0.0 | 0.0 | 0 | 0 | 0 | 4/5/2015 | 0.0 | 0.0 | 8.0 | 0 | 8 | 8 |
| 3/6/2013 | 0.0 | 0.0 | 0.0 | 0 | 0 | 0 | 4/6/2015 | 3.0 | 3.0 | 3.0 | 0 | 0 | 8 |
| 3/7/2013 | 0.0 | 0.0 | 0.0 | 0 | 0 | 0 | 4/7/2015 | 1.0 | 4.0 | 4.0 | 3 | 3 | 3 |
| 3/8/2013 | 0.0 | 0.0 | 0.0 | 0 | 0 | 0 | 4/8/2015 | 0.0 | 1.0 | 4.0 | 1 | 4 | 4 |
| 3/9/2013 | 13.5 | 13.5 | 13.5 | 0 | 0 | 0 | 4/9/2015 | 0.0 | 0.0 | 1.0 | 0 | 1 | 4 |
| 3/10/2013 | 0.5 | 14.0 | 14.0 | 0 | 0 | 0 | 4/10/2015 | 32.8 | 32.8 | 32.8 | 0 | 0 | 1 |
| 3/11/2013 | 0.0 | 0.5 | 14.0 | 14.2 | 14.2 | 14.2 | 4/11/2015 | 0.0 | 32.8 | 32.8 | 32.8 | 32.8 | 32.8 |
| 3/12/2013 | 0.0 | 0.0 | 0.5 | 0 | 14.2 | 14.2 | 4/12/2015 | 0.0 | 0.0 | 32.8 | 0 | 32.8 | 32.8 |
| 3/13/2013 | 0.0 | 0.0 | 0.0 | 0 | 0 | 14.2 | 4/13/2015 | 0.0 | 0.0 | 0.0 | 0 | 0 | 32.8 |
| 3/14/2013 | 0.0 | 0.0 | 0.0 | 0 | 0 | 0 | 4/14/2015 | 0.0 | 0.0 | 0.0 | 0 | 0 | 0 |
| 3/15/2013 | 0.0 | 0.0 | 0.0 | 0 | 0 | 0 | 4/15/2015 | 1.6 | 1.6 | 1.6 | 0 | 0 | 0 |
| 3/16/2013 | 0.0 | 0.0 | 0.0 | 0 | 0 | 0 | 4/16/2015 | 0.0 | 1.6 | 1.6 | 1.6 | 1.6 | 1.6 |
| 3/17/2013 | 7.6 | 7.6 | 7.6 | 0 | 0 | 0 | 4/17/2015 | 12.4 | 12.4 | 14.0 | 0 | 1.6 | 1.6 |
| 3/18/2013 | 2.1 | 9.7 | 9.7 | 0 | 0 | 0 | 4/18/2015 | 4.4 | 16.8 | 16.8 | 12.4 | 12.4 | 14 |
| 3/19/2013 | 0.0 | 2.1 | 9.7 | 12.4 | 12.4 | 12.4 | 4/19/2015 | 8.4 | 12.8 | 25.2 | 4.4 | 16.8 | 16.8 |
| 3/20/2013 | 0.0 | 0.0 | 2.1 | 2.7 | 15.1 | 15.1 | 4/20/2015 | 0.0 | 8.4 | 12.8 | 8.4 | 12.8 | 25.2 |
| 3/21/2013 | 0.0 | 0.0 | 0.0 | 0 | 2.7 | 15.1 | 4/21/2015 | 10.8 | 10.8 | 19.2 | 0 | 8.4 | 12.8 |
| 3/22/2013 | 0.0 | 0.0 | 0.0 | 0 | 0 | 2.7 | 4/22/2015 | 1.0 | 11.8 | 11.8 | 10.8 | 10.8 | 19.2 |
| 3/23/2013 | 0.0 | 0.0 | 0.0 | 0 | 0 | 0 | 4/23/2015 | 16.8 | 17.8 | 28.6 | 1 | 11.8 | 11.8 |
| 3/24/2013 | 0.0 | 0.0 | 0.0 | 0 | 0 | 0 | 4/24/2015 | 1.6 | 18.4 | 19.4 | 16.8 | 17.8 | 28.6 |
| 3/25/2013 | 0.0 | 0.0 | 0.0 | 0 | 0 | 0 | 4/25/2015 | 6.8 | 8.4 | 25.2 | 1.6 | 18.4 | 19.4 |
| 3/26/2013 | 14.8 | 14.8 | 14.8 | 0 | 0 | 0 | 4/26/2015 | 16.0 | 22.8 | 24.4 | 6.8 | 8.4 | 25.2 |
| 3/27/2013 | 2.7 | 17.5 | 17.5 | 0 | 0 | 0 | 4/27/2015 | 14.0 | 30.0 | 36.8 | 16 | 22.8 | 24.4 |
| 3/28/2013 | 0.0 | 2.7 | 17.5 | 18.5 | 18.5 | 18.5 | **4/28/2015** | **33.2** | **47.2** | **63.2** | **14** | **30** | **36.8** |
| 3/29/2013 | 27.5 | 27.5 | 30.2 | 0 | 18.5 | 18.5 | 4/29/2015 | 4.6 | 37.8 | 51.8 | 33.2 | 47.2 | 63.2 |
| **3/30/2013** | **33.6** | **61.1** | **61.1** | **0** | **0** | **18.5** | 4/30/2015 | 34.2 | 38.8 | 72.0 | 4.6 | 37.8 | 51.8 |
| 3/31/2013 | 0.0 | 33.6 | 61.1 | 29.8 | 29.8 | 29.8 | 5/1/2015 | 3.5 | 37.7 | 42.3 | 34.2 | 38.8 | 72 |
| 4/1/2013 | 1.1 | 1.1 | 34.7 | 0 | 29.8 | 29.8 | 5/2/2015 | 0.0 | 3.5 | 37.7 | 3.5 | 37.7 | 42.3 |
| 4/2/2013 | 0.0 | 1.1 | 1.1 | 0 | 0 | 29.8 | 5/3/2015 | 0.0 | 0.0 | 3.5 | 0 | 3.5 | 37.7 |
| 4/3/2013 | 0.0 | 0.0 | 1.1 | 0 | 0 | 0 | 5/4/2015 | 10.2 | 10.2 | 10.2 | 0 | 0 | 3.5 |
| 4/4/2013 | 6.9 | 6.9 | 6.9 | 7.8 | 7.8 | 7.8 | 5/5/2015 | 0.0 | 10.2 | 10.2 | 10.2 | 10.2 | 10.2 |
| 4/5/2013 | 4.0 | 10.9 | 10.9 | 6.6 | 14.4 | 14.4 | **5/6/2015** | **67.0** | **67.0** | **77.2** | **0** | **10.2** | **10.2** |
| 4/6/2013 | 7.8 | 11.8 | 18.7 | 40.5 | 47.1 | 54.9 | **5/7/2015** | **2.8** | **69.8** | **69.8** | **67** | **67** | **77.2** |
| 4/7/2013 | 23.0 | 30.8 | 34.8 | 60.5 | 101 | 107.6 | 5/8/2015 | 0.0 | 2.8 | 69.8 | 2.8 | 69.8 | 69.8 |
| 4/8/2013 | 14.0 | 37.0 | 44.8 | 19.2 | 79.7 | 120.2 | 5/9/2015 | 0.0 | 0.0 | 2.8 | 0 | 2.8 | 69.8 |
| 4/9/2013 | 15.6 | 29.6 | 52.6 | 12 | 31.2 | 91.7 | 5/10/2015 | 5.9 | 5.9 | 5.9 | 0 | 0 | 2.8 |
| 4/10/2013 | 0.0 | 15.6 | 29.6 | 0 | 12 | 31.2 | 5/11/2015 | 8.1 | 14.0 | 14.0 | 5.9 | 5.9 | 5.9 |
| 4/11/2013 | 1.3 | 1.3 | 16.9 | 0 | 0 | 12 | 5/12/2015 | 14.3 | 22.4 | 28.3 | 8.1 | 14 | 14 |
| 4/12/2013 | 30.8 | 32.1 | 32.1 | 57.7 | 57.7 | 57.7 | 5/13/2015 | 16.9 | 31.2 | 39.3 | 14.3 | 22.4 | 28.3 |
| 4/13/2013 | 11.8 | 42.6 | 43.9 | 16.8 | 74.5 | 74.5 | 5/14/2015 | 0.0 | 16.9 | 31.2 | 16.9 | 31.2 | 39.3 |
| 4/14/2013 | 6.8 | 18.6 | 49.4 | 6.4 | 23.2 | 80.9 | 5/15/2015 | 0.0 | 0.0 | 16.9 | 0 | 16.9 | 31.2 |
| 4/15/2013 | 0.0 | 6.8 | 18.6 | 0 | 6.4 | 23.2 | 5/16/2015 | 0.0 | 0.0 | 0.0 | 0 | 0 | 16.9 |
| 4/16/2013 | 27.8 | 27.8 | 34.6 | 27.6 | 27.6 | 34 | 5/17/2015 | 0.8 | 0.8 | 0.8 | 0 | 0 | 0 |
| 4/17/2013 | 0.0 | 27.8 | 27.8 | 0 | 27.6 | 27.6 | 5/18/2015 | 0.0 | 0.8 | 0.8 | 0.8 | 0.8 | 0.8 |
| **4/18/2013** | **24.4** | **24.4** | **52.2** | **23.2** | **23.2** | **50.8** | 5/19/2015 | 0.0 | 0.0 | 0.8 | 0 | 0.8 | 0.8 |
| 4/19/2013 | 7.7 | 32.1 | 32.1 | 7.9 | 31.1 | 31.1 | 5/20/2015 | 0.0 | 0.0 | 0.0 | 0 | 0 | 0.8 |
| 4/20/2013 | 0.0 | 7.7 | 32.1 | 0 | 7.9 | 31.1 | 5/21/2015 | 1.8 | 1.8 | 1.8 | 0 | 0 | 0 |
| 4/21/2013 | 1.0 | 1.0 | 8.7 | 0 | 0 | 7.9 | 5/22/2015 | 1.0 | 2.8 | 2.8 | 1.8 | 1.8 | 1.8 |
| 4/22/2013 | 14.3 | 15.3 | 15.3 | 21.9 | 21.9 | 21.9 | 5/23/2015 | 0.0 | 1.0 | 2.8 | 1 | 2.8 | 2.8 |
| 4/23/2013 | 4.3 | 18.6 | 19.6 | 1.7 | 23.6 | 23.6 | 5/24/2015 | 0.0 | 0.0 | 1.0 | 0 | 1 | 2.8 |
| 4/24/2013 | 1.4 | 5.7 | 20.0 | 0 | 1.7 | 23.6 | 5/25/2015 | 0.0 | 0.0 | 0.0 | 0 | 0 | 1 |
| 4/25/2013 | 0.0 | 1.4 | 5.7 | 0 | 0 | 1.7 | 5/26/2015 | 0.0 | 0.0 | 0.0 | 0 | 0 | 0 |
| 4/26/2013 | 0.0 | 0.0 | 1.4 | 0 | 0 | 0 | 5/27/2015 | 0.0 | 0.0 | 0.0 | 0 | 0 | 0 |
| 4/27/2013 | 0.0 | 0.0 | 0.0 | 0 | 0 | 0 | 5/28/2015 | 0.0 | 0.0 | 0.0 | 0 | 0 | 0 |
| 4/28/2013 | 36.0 | 36.0 | 36.0 | 39.5 | 39.5 | 39.5 | 5/29/2015 | 0.5 | 0.5 | 0.5 | 0 | 0 | 0 |
| 4/29/2013 | 1.7 | 37.7 | 37.7 | 0 | 39.5 | 39.5 | 5/30/2015 | 0.8 | 1.3 | 1.3 | 0.5 | 0.5 | 0.5 |
| 4/30/2013 | 3.6 | 5.3 | 41.3 | 0 | 0 | 39.5 | 5/31/2015 | 0.0 | 0.8 | 1.3 | 0.8 | 1.3 | 1.3 |

In the above table, the date on which daily maximum rainfall was recorded for each sifnificant flood event are highlited.

The return period of the significant flood events was computed, and Table 6 below shows the corresponding return period of the significant flood event that occurred in the town. For example, the flood that occurred in January 1993 had a return period of six years, with a corresponding amount of rainfall equal to 81.7 mm recorded on the 20th January 1993.

**Table 6.** Summary table of the return period corresponding to the major flood event.

| Date | Daily Rainfall in mm | Forecast T in Years |
|---|---|---|
| 1/20/1993 | 81.7 | 6 |
| 3/30/2013 | 33.6 | 1 |
| 4/18/2013 | 24.4 | 1 |
| 4/28/2015 | 33.2 | 1 |

## 4. Conclusions

Rainfall intensity can vary depending on the geographical location, and the study of rainfall variation over the years has emerged as an important aspect of flood management. However, an important consideration in flood frequency analysis is the determination of the best-fitted distribution in order to compute the appropriate return period. In this study, different probability distribution functions were applied to the time series data of two stations in Narok Town. The analysis covered not only the maximum daily data, but also the maximum two and three days of rainfall. This was done because it was noticed that some days reported flooding even when daily rainfall was not much, owing to the fact that the soil had absorbs the rainfall in the previous days, and so even a small amount of additional rain led to flooding. The K–D test and Chi-square test were applied to each distribution and the values of each parameter helped to determine the best-fitted distribution in each case. This was followed by the computation of the 5, 10, 25, 50, and 100 years return periods rainfall. The results of the statistical analysis were used together with PGIS to compute the corresponding return period of significant flood events that occurred in the town. Findings from the study show that integrating PGIS in flood management could be helpful in gathering information on past flood events. This opens up the potential for extending application of PGIS into the analysis of flood extent and flood depth mapping. Going forward, the methods used in this study could be applied in developing flood hazard maps for each maximum rainfall intensity, while results of the study could be used by the government in flood management in Kenya. More significantly, the method and results could be used by development planners at the county and national levels in identifying flood-prone areas in Narok and in developing mitigation strategies. The outcome of the study could also be used in urban planning for Narok Town.

**Author Contributions:** E.A.Y.H.-D. carried out the research, J.M.G., M.N. and Z.A.G. supervised the research.

**Funding:** The authors acknowledge the financial support from the African Union and Pan African University-Institute for Basic Sciences Technology and Innovation, Kenya.

**Conflicts of Interest:** The authors declare no conflict of interest.

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
