# Peer review of "Flood Frequency Analysis Using Participatory GIS and Rainfall Data for Two Stations in Narok Town, Kenya"

_hydrology, doi:10.3390/hydrology6040090_

Round 1

Reviewer 1 Report

General comments

For citing the references follow the Instruction for authors: In the text, reference numbers should be placed in square brackets [ ], and placed before the punctuation; for example [1], [1–3] or [1,3].

Specific comments and corrections:

Line 41: Oosterban (2019)… see the General comments

Line 51: … which indicates

Line 55: references: see the General comments

Lines 79-80: you state that the soil group is C and the curve number is 86. Please add references and/or related studies

Lines 80-81: Please specify here that WS is for Water supply station and MS is for Meteorological station, used hereafter in the text

Lines 97-104: references: see the General comments

Line 124: Typo error: close the bracket after Leuven

Lines 123-129: Please add references

Line 134: Please check the sentence, there are grammatical errors

Lines 142- 162: Step 3: it is not necessary to illustrate all the methods that could be used but only the methods chosen for you analysis and why

Lines 164-171: I suppose that all the readers of this journal know very well what “return period” means

Line 178: you state that the distribution of rain is homogeneous in the catchment and then in line 182 that the rainfall varies along (or among?)  the stations. The statements are contradictory.

Line 183: 1975 and 2015

Line 186: in the Figure 2 please improve the description of ordinate and the caption with Series 1 and 2

Line 194: Typo error: and chi-square, are highlighted

Line 190-211: please improve the description of the parameters, also in the tables 3 and 4

Line 199: goodness-of-fit test

Line 215: Table 5 and not 4

Line 218: see above

Line 228: Typo error: floods

Line 229: Typo error: events

Line 230: The following graphs? Please specify/change

Line 236: Table 6 and not 5

Line 241: see above

Line 236: There is a need to notice

Line 244: Table 7 and not 6

Line 248: see above

Line 257: Typo error: … daily rainfall was

Author Response

We would like to thank the reviewer for reading our work and appreciate that he / she has found the research method is adequately described and the result well presented. The document has been revised with regard to English spelling. The section below highlights our response to the reviewer's comments

Reviewer 2 Report

In this manuscript, the authors presented the results of their PGIS model. In addition to the 6 comments or queries added in the marked copy. My main suggest is that the authors pointed out the advantage or strength of using PGIS in the prediction flood frequency. However, their only presented their prediction. I would suggest the authors present a comparison between the proposed PGIS and the conventional GIS models.  

Another minor comment is: some information of references are missing. Please cross-check before resubmission.

Author Response

We would like to thank the reviewer for reading our work. The English spelling document has been revised. The following shows our reply to remarks from the reviewer.

Round 2

Reviewer 1 Report

No new comments or suggestions

Reviewer 2 Report

The authors have addressed my comments from the preivous review. I recommend it now.